# Macrocyclic peptides exhibit antiviral effects against influenza virus HA and prevent pneumonia in animal models

Makoto Saito[1,9], Yasushi Itoh[2,9], Fumihiko Yasui[1,9], Tsubasa Munakata[1], Daisuke Yamane [1], Makoto Ozawa [3], Risa Ito[4], Takayuki Katoh [4], Hirohito Ishigaki [2], Misako Nakayama[2], Shintaro Shichinohe[2], Kenzaburo Yamaji[1], Naoki Yamamoto[1], Ai Ikejiri[1], Tomoko Honda[1], Takahiro Sanada[1], Yoshihiro Sakoda [5], Hiroshi Kida[6], Thi Quynh Mai Le[7], Yoshihiro Kawaoka [8], Kazumasa Ogasawara[2], Kyoko Tsukiyama-Kohara[3✉], Hiroaki Suga [4✉] & Michinori Kohara [1✉]

Most anti-influenza drugs currently used, such as oseltamivir and zanamivir, inhibit the enzymatic activity of neuraminidase. However, neuraminidase inhibitor-resistant viruses have already been identified from various influenza virus isolates. Here, we report the development of a class of macrocyclic peptides that bind the influenza viral envelope protein hemagglutinin, named iHA. Of 28 iHAs examined, iHA-24 and iHA-100 have inhibitory effects on the in vitro replication of a wide range of Group 1 influenza viruses. In particular, iHA-100 bifunctionally inhibits hemagglutinin-mediated adsorption and membrane fusion through binding to the stalk domain of hemagglutinin. Moreover, iHA-100 shows powerful efficacy in inhibiting the growth of highly pathogenic influenza viruses and preventing severe pneumonia at later stages of infection in mouse and non-human primate cynomolgus macaque models. This study shows the potential for developing cyclic peptides that can be produced more efficiently than antibodies and have multiple functions as next-generation, mid-sized biomolecules.

[1] Department of Microbiology and Cell Biology, Tokyo Metropolitan Institute of Medical Science, Setagaya-ku, Tokyo, Japan. [2] Division of Pathogenesis and Disease Regulation, Department of Pathology, Shiga University of Medical Science, Setatsukinowa, Otsu, Shiga, Japan. [3] Transboundary Animal Diseases Center, Joint Faculty of Veterinary Medicine, Kagoshima University, Kagoshima, Japan. [4] Department of Chemistry, Graduate School of Science, The University of Tokyo, Bunkyo-ku, Tokyo, Japan. [5] Laboratory of Microbiology, Faculty of Veterinary Medicine, Hokkaido University, Sapporo, Japan. [6] Hokkaido University Research Center for Zoonosis Control, Sapporo, Japan. [7] National Institute of Hygiene and Epidemiology, Hanoi, Vietnam. [8] Division of Virology, Department of Microbiology and Immunology, The Institute of Medical Science, The University of Tokyo, Tokyo, Japan. [9] These authors contributed equally: Makoto Saito, Yasushi Itoh, Fumihiko Yasui. ✉email: kkohara@vet.kagoshima-u.ac.jp; hsuga@chem.s.u-tokyo.ac.jp; kohara-mc@igakuken.or.jp

The influenza A virus is a great public health issue[1]. Human H1N1 and H3N2 viruses are highly contagious and cause respiratory disorders with high morbidity and mortality[2]. In addition, highly pathogenic H5N1 avian influenza viruses sporadically transmit to humans with a high fatality rate. Although neuraminidase (NA) inhibitors are globally approved for influenza treatment, NA inhibitor-resistant viruses have already been isolated among seasonal H1N1[3,4], pandemic (H1N1) 2009[5–7], and even highly pathogenic avian H5N1[8,9] viruses. These resistant viruses can only be controlled by an alternative antiviral with a mechanism of action that is totally different from NA inhibition.

Hemagglutinin (HA), an envelope protein of the influenza virus that forms trimers on the surface of the virions, plays a critical role in the virus replication cycle. HA mediates cell attachment through binding to receptor molecules and low pH-dependent membrane fusion in endosomes[10]. Although HA is classified into 16 subtypes based on its antigenicity, all HA molecules share the same fundamental biological function[10]. Since HA is the most immunogenic among influenza viral proteins and HA-targeted antibodies have the potential to neutralize HA-mediated virion-cell binding[10], several HA-specific monoclonal antibodies have been established and demonstrated to be useful as anti-influenza drugs. However, the antibody binding sites on HA are frequently mutated, and thus escape mutants are easily generated. Moreover, the molecular size of antibodies is comparable to that of the HA trimer, thus limiting access to the HA functional domain.

An emerging screening technology, the Random non-standard Peptides Integrated Discovery (RaPID) system, is constructed by combining mRNA display technology with the Flexible In vitro Translation (FIT) system, which consists of the artificial ribozyme "flexizyme" and an *Escherichia coli*-reconstituted cell-free translation system. This technology allows us to freely express thioether-macrocyclic peptides incorporating non-standard amino acids in a custom-made in vitro translation system and display this massive library (greater than $10^{12}$ members) on cognate mRNA templates. The recent development of the RaPID system enables the identification of high-affinity ligands for a protein of interest from easily prepared libraries consisting of over a trillion macrocyclic peptides[11–13].

In this work, to devise smaller molecules capable of binding to influenza viral HA as potential antiviral agents[14] (Supplementary Fig. 1a), we use the RaPID system to obtain HA-targeting macrocycles, named iHAs. Among candidate iHAs, iHA-24 and iHA-100 show inhibitory effects on the in vitro replication of a wide range of Group 1 influenza viruses. Notably, iHA-100 bifunctionally inhibited viral adsorption and membrane fusion, both of which are mediated by HA. The binding site for iHA-100 in HA is the stalk domain, which is highly conserved among various subtypes of the influenza virus. Moreover, iHA-100 exhibits efficacy in inhibiting virus growth and preventing severe pneumonia at later stages of infection in mouse and non-human primate cynomolgus macaque models.

## Results

### Macrocycles iHA-100 and iHA-24 inhibit Group 1 influenza A viruses.
In the RaPID system, iterative selection rounds were performed to enrich potent macrocyclic binders against recombinant HA derived from the highly pathogenic avian influenza virus A/Vietnam/1203/04 (H5N1/VietNam1203; clade 1) or A/Bar-Headed goose/Qinghai Lake/1 A/05 (H5N1/Qinghai Lake: clade 2.2) (Supplementary Fig. 1b). We utilized two libraries of thioether-macrocycles in which the random region consists of only proteinogenic amino acids or both 11 proteinogenic and four types of N-methyl-amino acids of choice[15]. The latter library was designed to assure greater metabolic stability than the former type with simple thioether-macrocyclic peptides. After five rounds of selection, we found an appreciable increase in the cDNA recovery rate. Sequencing of 69 molecular clones from the selected cDNAs revealed 28 candidates for an inhibitor of HA (iHA) (Supplementary Fig. 1c). Twenty-four iHA candidates (iHA-1–24) and four iHA candidates (iHA-100–103) were identified against the HA protein of H5N1/VietNam1203 and H5N1/Qinghai Lake, respectively. Of the 28 candidates, eight iHA macrocycles (iHA-11, −12, −14, −18, −19, −23, −24, and −100) (Fig. 1a) inhibited plaque formation of the low pathogenic H5N1 avian virus A/duck/Hokkaido/Vac-3/07 (H5N1/Vac-3) (Fig. 1b, left panel).

To determine if the iHA macrocycles bound to influenza viruses, we established a pull-down assay for the quantification of peptide-associated viral RNAs. Higher copy numbers of the viral RNA were detected in the avidin-biotin complex with each of the eight iHAs tested compared to a control biotinylated peptide (Fig. 1b, right panel). Thus, iHAs potently bind to not only recombinant HA but also influenza virions, indicating that the nature of this inhibitory activity of iHAs is their strong binding to HA.

Among these macrocycles, iHA-24 and iHA-100 (structures are shown in Fig. 1c) remarkably reduced the plaque numbers of H5N1/Vac3 (Fig. 1d) and their sizes (Supplementary Fig. 2), and also inhibited plaque formation of different antigenicities among clades of H5N1 viruses: A/Vietnam/UT3040/04 (H5N1/Vietnam; clade 1), A/whooper swan/Mongolia/3/05 (H5N1/Mongolia; clade 2.2), and A/whooper swan/Hokkaido/1/08 (H5N1/Hokkaido; clade 2.3.2.1) (Fig. 1e-g). Intriguingly, iHA-24 and iHA-100 were also more effective against the H1N1 virus laboratory strain, A/Puerto Rico/8/34 (H1N1/PR8), the pandemic (H1N1) 2009 virus A/Tokyo/2619/09 (H1N1/Tokyo2619), and A/Narita/1/09 (H1N1/Narita) than zanamivir (Fig. 1h-j). Moreover, iHA-24 and iHA-100 inhibited the H2N2 virus (Fig. 1k, Supplementary Table 1). HAs of H1N1, H2N2, and H5N1 viruses belong to the Group 1 HA subtype. Consistent with these data, iHA-24 and iHA-100 bound to HA on H1N1 and H2N2 virions (Supplementary Fig. 3). Thus, iHA-24 and iHA-100 inhibited the replication of a wide range of Group 1 influenza viruses via interaction with HAs.

### iHA-100 bifunctionally inhibits viral adsorption and membrane fusion through binding to the stalk domain of HA.
Viral replication steps including adsorption, endocytosis, fusion, and uncoating are required for influenza A virus invasion of host cells[16]. To investigate the mechanism of the neutralizing activity of iHA-100, we examined whether iHA-100 actually interferes with the adsorption of the virus to host cells. H5N1/Vac-3 was pre-treated with iHA-100 and then adsorbed to Madin-Darby canine kidney (MDCK) cells on ice to prevent endocytosis. The virus-adsorbed cells were extensively washed, and residual viral RNA was measured. iHA-100 inhibited the adsorption of H5N1/Vac-3 in a dose-dependent manner (Fig. 2a, IC$_{50}$ = 0.036 μM). Moreover, the $K_D$ value of iHA-100 binding to recombinant HA protein was estimated by surface plasmon resonance (SPR) to be 1.5 nM (Fig. 2b, Supplementary Fig. 4).

To further examine the target of iHA-100 after adsorption, iHA-100 was applied at the virus infection steps of uncoating and fusion (Fig. 2c). Virus replication was inhibited by iHA-100 treatment at −1 h, 0 h, 0.3 h, 1 h, and 1.5 h after infection, but uncoating at 3 h after infection was not inhibited (Fig. 2d). The negative control peptide iHA-12sf, which has a scrambled iHA-12 amino acid sequence (see "Methods" section for details), did not inhibit virus entry. Thus, the fusion step may be a critical target of

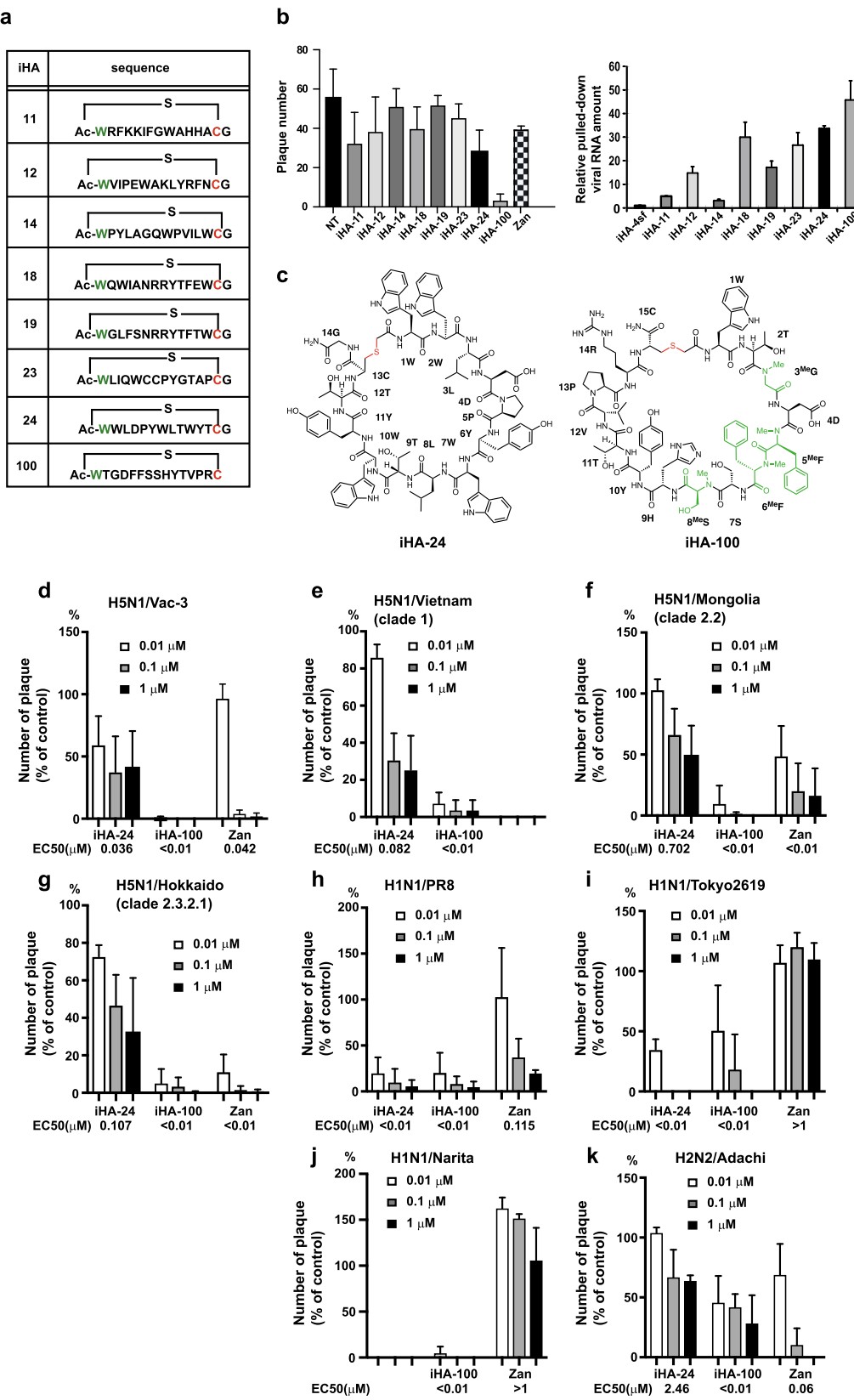

iHA-100. Furthermore, MDCK cells infected with H5N1/Vac-3 at a multiplicity of infection of 0.1 were treated with iHA-100 3 h after infection in the absence (Fig. 2e) or presence (Fig. 2f) of trypsin. Virus spread occurs in an HA-dependent manner in the presence of trypsin, but no virus spread is observed in the absence of trypsin. As shown in Fig. 2f, treatment with iHA-100 3 h after

infection completely inhibited viral RNA replication as in the absence of trypsin (Fig. 2e), indicating inhibition of viral spread.

To characterize the inhibition of viral fusion by iHA-100, we tested its effect on HA-mediated membrane fusion activity using the polykaryon formation assay. The membrane fusion activity was assessed by counting polykaryon numbers. Polykaryon

**Fig. 1 Macrocycles iHA-100 and iHA-24 inhibit Group 1 influenza A viruses. a** HA-targeted macrocycles selected with the RaPID system. Amino acid sequences of the selected HA-binding macrocycle candidates are shown. Images of protein molecules were created using the software program PyMOL. **b** (left panel), Eight iHA macrocycles (iHA-11, −12, −14, −18, −19, −23, −24, and −100, NT; without iHA) inhibited plaque formation of the low pathogenic H5N1 avian virus A/duck/Hokkaido/Vac-3/07 (H5N1/Vac-3). (right panel), Binding of the iHAs to influenza viruses. RNAs were extracted from A/duck/Hokkaido/Vac-3/07 (H5N1/Vac-3) virus bound to biotin-iHAs and subjected to real-time PCR to quantify viral RNAs according to the M gene. Relative RNA amounts associated with each biotin-iHA to that of the iHA-4sf control macrocycle, which consists of a scrambled iHA-4 amino acid sequence, are shown. Error bars indicate standard deviations (SD) of three replicates, and the results showed are representative of two biologically independent experiments. **c** Chemical structure of iHA-24 (left) and iHA-100 (right) macrocycles. **d–k** Effect of the selected macrocycles on plaque formation by Group 1 influenza viruses. Percentage (%) of plaque reduction normalized to the untreated control is indicated. Results are mean values from three biologically independent experiments, and error bars indicate the SD of three replicates. Viral strains are H5N1/Vac-3 (clade unclassified) (**d**), H5N1/ Vietnam (clade 1) (**e**), H5N1/Mongolia (clade 2.2) (**f**), H5N1/Hokkaido (clade 2.3.2.1) (**g**), H1N1/PR8 (**h**), pandemic (H1N1) 2009 viruses H1N1/Tokyo2619 (**i**), H1N1/Narita (**j**), and H2N2/Adachi (**k**). MDCK cells were infected with 100 PFU of each virus that was pre-treated with 0.01, 0.1, or 1 μM iHA-24 or iHA-100. At 1 h post-infection, cells were washed and overlaid with agarose medium containing 0.01, 0.1, or 1 μM of either the indicated macrocycle or zanamivir. After 48–72 h of incubation, plaque numbers were counted and divided by their back-titration counterparts (i.e., plaque numbers formed by untreated viruses in agarose medium without the macrocycle or zanamivir).

---

formation mediated by H5 and H1 HAs was significantly inhibited by iHA-100 in a dose-dependent manner (Fig. 2g, and Supplementary Fig. 5). We further confirmed the inhibitory effect of iHA-100 on membrane fusion with a red blood cell (RBC) fusion assay (Supplementary Fig. 6). Thus, iHA-100 inhibited HA-mediated membrane fusion. Taken together, iHA-100 shows potent neutralization activity by inhibiting HA-mediated adsorption, fusion, and virus spread (Fig. 2a, d, f, g).

To map potential binding site(s) of macrocycles to HA, we isolated escape mutants (Fig. 2h, Supplementary Fig. 7). After three serial passages of H1N1/PR8 or H5N1/Vac-3 in cultured cells in the presence of iHA-100, we isolated 10 clones that formed plaques in the presence of 0.1 μM iHA-100 more efficiently than the parental virus. The HA gene sequences of these escape mutants were determined. Escape mutations against iHA-100 were located in the stalk domain (Fig. 2h, Supplementary Fig. 7a), which is highly conserved between Group 1 HAs. In addition, we confirmed that these mutations were responsible for the reduced sensitivity to iHA-100 (Supplementary Fig. 7b, c). These substitutions may change the conformation of the stalk domain, decreasing the accessibility of iHA-100.

To gain initial insight into possible stalk binding of iHA-100, we examined whether iHA-100 can protect HA protein from pH-induced protease degradation. Stalk-targeting Abs like CR6261 stabilize the prefusion conformation and block HA conformational change at low pH and subsequent susceptibility to trypsin[17,18]. In our trypsin susceptibility assay, HA was normally degraded by trypsin completely at acidic pH and partially at neutral pH, but when incubated with iHA-100 or CR6261, HA became protease-resistant (Fig. 2i), suggesting binding of iHA-100 to the HA stalk region. These results also indicated that iHA-100 protected HA from trypsin cleavage at neutral pH (Fig. 2i, compare lanes 4 and 6). We consider that iHA-100 primarily interacted with the stalk domain to interfere with the conformational change in the HA protein at neutral conditions to block the viral adsorption process. Collectively, iHA-100 inhibited the adsorption and fusion step of the influenza virus by binding to the stalk domain of HA.

**Efficacy of iHA-100 against lethal influenza virus infection in mice.** We evaluated the in vivo efficacy of iHA-100 against influenza virus infection in a murine lethal infection model (Fig. 3). H5N1-infected mice were intranasally administered 1.9 mg/kg/day iHA-100 at 0 (3 h), 2, 4, or 6 days after infection. When the administration was started 0 days after H5N1 infection, bodyweight showed more fluctuation in mice treated with iHA-100 compared to the zanamivir-treated group. The survival rate in the iHA-100-treated group was comparable with the survival rate in the zanamivir-treated group (Fig. 3a, b). Although all mice in the zanamivir-treated group died due to severe weight loss when the administration was delayed to 4–8 days post-infection (dpi) and 6–10 dpi (Fig. 3a, b), the iHA-100-treated group exhibited slight weight loss, eventually recovered, and showed 40% survival. In both early (0–4 dpi) and delayed (4–8 dpi) administration of iHA-100, the virus-related increase in lung weight and viral replication in the lungs were significantly suppressed (Fig. 3c–f). In addition, the administration of iHA-100 prevented the progression of pneumonia (Fig. 3g). Thus, iHA-100 exhibited a high efficacy whether administered during the early phase (after 0 or 2 days) or late phase (after 4 or 6 days) of H5N1 infection.

**Efficacy of iHA-100 against highly pathogenic H5N1 in a non-human primate infection model.** We further evaluated the in vivo efficacy of iHA-100 against influenza virus infection in a non-human primate cynomolgus macaque infection model (Fig. 4). Cynomolgus monkeys were intratracheally, intranasally, and orally infected with a total of $3 \times 10^6$ plaque-forming units (PFU)/monkey of H5N1/Vietnam, and then were intratracheally given 3 mg/kg/day iHA-100 once daily for 5 days beginning at 2 dpi. Immediately after infection (1 and 2 dpi), all monkeys exhibited increases of 3 °C in body temperature (around 40 °C) at night (Fig. 4a, b). The high body temperature in monkeys of the vehicle-treated group continued during the experimental period (Fig. 4a), resulting in the death of ID #1627 at 4 dpi. In contrast, the body temperature in monkeys in the iHA-100-treated group (ID #1597 and #1602) returned to a normal range (within 1 °C before infection, 37.5–38.5 °C) at 5–7 dpi (Fig. 4b). Although the fever continued in ID #1605, this may have been due to suppurative inflammation of the telemetry implantation site. The effect of treatment with iHA-100 on body temperature was plotted as the total body temperature increase from day 3 to day 7 (AUC day 3–7) (Fig. 4c). The average body temperature increase in the iHA-100-treated group was lower than that in the vehicle group, except for ID #1627 because of a low body temperature attributed to a serious clinical condition. Bodyweight loss was relatively attenuated in monkeys in the iHA-100-treated group compared to the two vehicle-treated monkeys (#1627 and #1635) (Fig. 4d). Regarding pathological findings, lung weights were relatively lower in the iHA-100-treated group than in the vehicle-treated group (Fig. 4e, $p = 0.062$). In nasal, oral, and bronchial swabs of vehicle-treated monkeys, the virus was detected until 6 dpi. In contrast, the virus in these swabs of iHA-100-treated monkeys disappeared at 3–6 dpi (Fig. 4f, g, h). By calculating the area under the curve, the iHA-100-treated group showed significantly lower virus shedding after treatment

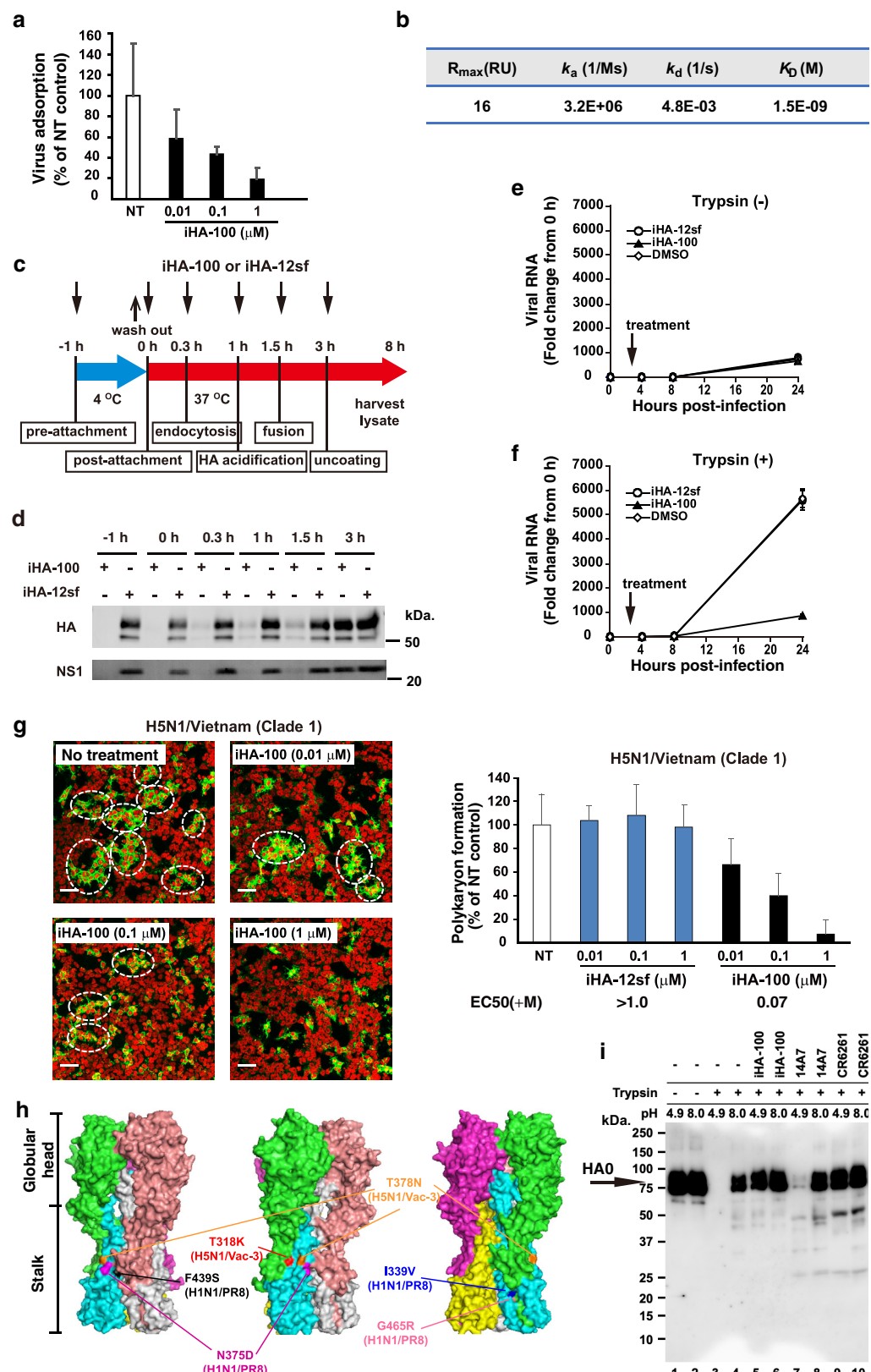

(3–7 dpi) in nasal swabs, oral swabs, and bronchus brush (Supplementary Fig. 8). Virological analysis of autopsy lung samples at 7 dpi showed low virus titers in the lungs of two monkeys (#1605 and #1602) that were treated with iHA-100 (Fig. 4i). In contrast, the virus was detected in almost all lung lobes in vehicle-treated monkeys (Fig. 4i). Macroscopic findings of the extirpated lungs of

the vehicle-treated group indicated obvious inflammatory lesions compared with the lungs of the iHA-100-treated group (Supplementary Fig. 9).

To investigate the influence of iHA-100 on the pathology of H5N1 infection, a comprehensive analysis of cytokines was performed. Inflammatory cytokines (interferon (IFN)-γ and

**Fig. 2 In vitro characterization of iHA-100.** Results are shown as mean values and are representative of three biologically independent experiments. Error bars indicate SD of three replicates. **a** Effect of iHA-100 on the adsorption of H5N1 to host cells. H5N1/Vac-3 ($2.5 \times 10^4$ PFU) is pre-treated with various concentrations of iHA-100 (0.01, 0.1, or 1 μM) for 1 h and then adsorbed to MDCK cells ($5 \times 10^4$ cells) on ice for 1 h. The virus-adsorbed cells are extensively washed and used for qRT-PCR. Virus adsorption is represented as the relative value of a nontreated virus. **b** Single-cycle kinetic analysis of the interaction between HA and iHA-100 by surface plasmon resonance (SPR). SPR kinetic data of iHA-100 binding to H5 HA is shown. **c, d** Time course of the addition of iHA-100. H5N1/Vac-3 was pre-treated with the macrocycles (−1 h; pre-attachment), or virus-adsorbed cells were added at 0 h (post-attachment), 20 min (endocytosis), 1 h (HA acidification), 1.5 h (fusion), or 3 h (uncoating) after infection. At 8 h after infection, infected cells were lysed and used for western blotting. **e, f** Effect of iHA-100 on virus spread. MDCK cells were infected with H5N1/Vac-3 at a multiplicity of infection of 0.1 and treated with iHA-100 at 3 h after infection in the absence (**e**) or presence (**f**) of trypsin. The absence of trypsin resulted in complete inhibition of virus spread. At 0, 4, 8, and 24 h after infection, RNAs were prepared from infected cells and used for qRT-PCR. **g** Effect of iHA-100 on influenza viral HA-mediated polykaryon formation. HEK293 cells transiently expressing HA from A/Vietnam/1194/04 (H5N1; clade 1) were trypsinized and exposed to low pH. After incubation at 37 °C for 6 h, HA (green in left images) and nuclei (red in left images) were visualized with immunofluorescence staining. The polykaryons are indicated by dotted ovals. Right panel: Percentage (%) of polykaryon formation normalized to the no treatment (NT) control was indicated. Vertical error bars indicate the SD of five randomly selected fields (**g**). iHA-12sf is a control macrocycle that consists of a scrambled iHA-12 amino acid sequence. **h,** Structure of HA and escape mutants. HA mutations found in escape mutants from iHA-100 are indicated in a schematic diagram of the three-dimensional structures of the HA trimer (created by PyMOL software). **i** iHA-100 and anti-HA stalk Ab protected HA from low pH-induced protease sensitivity. Exposure to low pH rendered H5 HA sensitive to trypsin digestion, but iHA-100 prevented conversion to the protease-susceptible conformation. 14A7 and CR6261 are anti-HA head and stalk antibodies, respectively. CR6261 functioned as a positive control in the trypsin susceptibility assays. HA was detected in immunoblots with rabbit anti-H5 HA antibody.

interleukin (IL)-6 in the lungs of iHA-100-treated monkeys were significantly reduced compared with vehicle-treated monkeys (Fig. 4j, k). In addition, IL-15 was also decreased in the lungs of iHA-100-administered monkeys (Fig. 4l), but other cytokines did not show significant changes (Supplementary Fig. 10). IL-15 is involved in the pathogenesis of influenza virus-induced acute lung injury[19]. These findings suggested that H5N1-induced inflammation and pathogenesis were suppressed by the administration of iHA-100. Thus, the results of the non-human primate cynomolgus macaque experiment support the results of the mouse experiments and are proof of concept design. We also assessed the stability of iHA-100 in serum and showed that its half-life is approximately 3.85 h (Supplementary Fig. 11). Taken together, our data suggest that iHA-100 is a candidate antiviral agent that inhibits both virus replication and pathogenesis in vivo.

## Discussion

Hemagglutinin (HA) contributes to the binding of the influenza virus to its receptors and is known as the main target for neutralizing antibodies. However, HA shows antigenic diversity due to high-frequency mutation; therefore, the effectiveness of neutralizing antibodies targeting HA is limited to specific strains of the virus. In recent years, broadly neutralizing antibodies (bNAb) that have a different mechanism of action from conventional neutralizing antibodies have been reported to inhibit the infection of a wide range of subtypes[17,18]. In this study, 28 HA-targeting macrocyclic peptides (iHAs) were found using the RaPID system. Of this iHAs, iHA-100 and iHA-24 effectively inhibited the in vitro replication of various Group 1 subtypes (Fig. 1d-k).

One of the mechanisms of action of the antiviral effect exerted by iHA-100 is inhibition of HA-mediated membrane fusion (Fig. 2g, Supplementary Figs. 5 and 6), similar to bNAb. In HA protein, the stalk domain is highly conserved among subtypes, as opposed to the globular head domain, which has diverse antigenicity. By binding to the stalk domain (Fig. 2h, and Supplementary Fig. 7), iHA-100 can inhibit HA-mediated membrane fusion of a wide range of subtype viruses, resulting in the prevention of viral entry into host cells. Recently, it has been reported that the influenza virus spreads by an alternative infection mode (cell-to-cell transmission) that does not require NA activity-dependent viral release, and thus shows NA inhibitor resistance[20]. Cell-to-cell transmission requires trypsin-induced HA maturation, which leads to viral entry into adjacent cells

dependent on mature HA-mediated membrane fusion. Therefore, iHA-100, which has inhibitory activity against HA-mediated membrane fusion, may also inhibit the cell-to-cell transmission of influenza viruses that cannot be inhibited by conventional neutralizing antibodies and NA inhibitors[20].

Another mechanism of action of the antiviral effect exerted by iHA-100 is inhibition of viral adsorption to host cells (Fig. 2a). Since the globular head domain, not the stalk domain, is involved in viral adsorption (binding to the receptor), bNAbs that bind the stalk domain do not have this mechanism of action. Although iHA-100 also binds to the stalk domain like bNAbs, the exact mechanism of how it exhibits inhibition of virus adsorption is unclear at present. Intriguingly, in the presence of iHA-100, trypsin-induced cleavage of HA0 was inhibited under not only acidic conditions but also neutral conditions (Fig. 2i). This indicates that iHA-100 prevents trypsin cleavage of HA0 on the cell surface (neutral conditions). We consider that iHA-100 primarily interacts with the stalk domain to interfere with the cleavage of HA0 protein in neutral conditions to block the viral adsorption process. Alternatively, in the analysis of escape mutations against iHA-100, mutations in the stalk domain were mainly found (Fig. 2h), while a mutation in the globular head domain (E219G) was also identified (Supplementary Fig. 7c). Therefore, iHA-100 mainly binds to the stalk domain, but it might also bind to the globular head domain. The 219th residue of HA constitutes the 220-loop involved in receptor binding[21] and has been reported to affect virus binding to the human receptor[22]. We consider that iHA-100 may inhibit the binding of HA to the receptor (virus adsorption) through binding to the globular head domain.

Furthermore, iHA-100 can act on HA intracellularly even if it is applied after virus entry (Fig. 2c–f). It has been reported that HA binds to STING and INFAR1, resulting in suppression of innate immune response signaling[23,24]. Thus, iHA-100 can bind to intracellular HA and might inhibit interactions among HA, STING, and INFAR1, thereby canceling the HA-mediated blockade of the innate immune signal. Therefore, by targeting HA, iHA-100 can inhibit virus entry (direct inhibition of infection), and it is also expected to cancel the suppression of innate immunity mediated by HA (indirect inhibition of infection). In animal models, viral growth was significantly suppressed and the progress of pulmonary edema/pneumonia ceased in the iHA administration group (Figs. 3 and 4). Aberrant innate immune responses are considered to be a determinant of severe

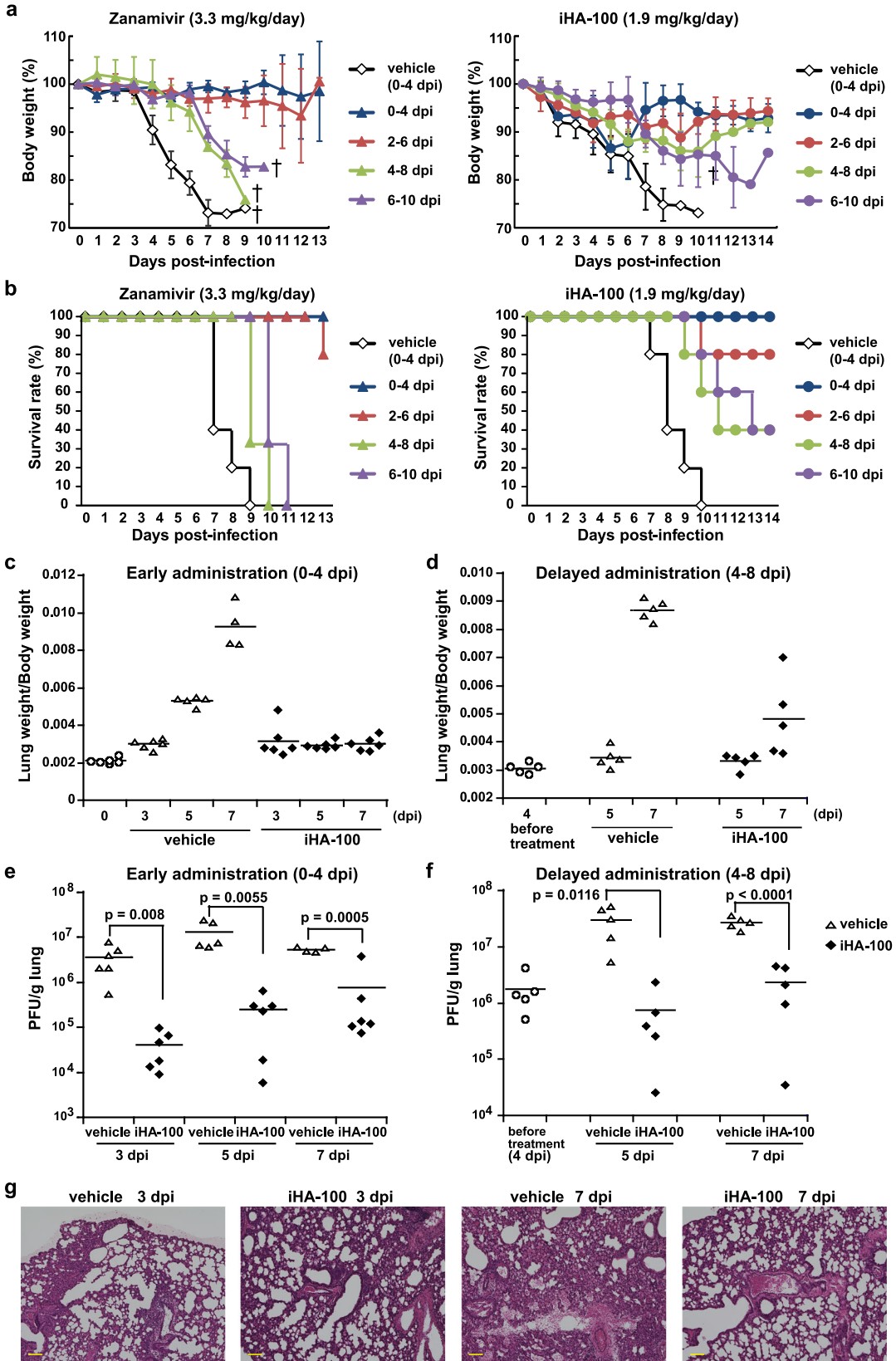

pathogenicity caused by influenza virus infection[25]. Suppression of pathogenesis as seen in the iHA-100 administration group might have been the result of iHA-mediated cancellation of innate immune response suppression caused by influenza virus infection. These effects of iHA-100 may contribute to its anti-viral effect in the late stages of influenza virus infection in vivo.

This study suggests that macrocyclic peptides may be powerful molecules to target the HA stalk region and may lead to the development of broad-spectrum influenza virus inhibitors, despite the considerable difference in antigenicity among virus strains. The RaPID system can attain a broader anti-viral spectrum of macrocycles. Thus, building on this work, we expect that

**Fig. 3 Efficacy of iHA-100 against lethal influenza virus infection in mice.** Five mice per group were intranasally inoculated with 5 $MLD_{50}$ of A/whooper swan/Hokkaido/1/08 (H5N1) and were then intranasally administered iHA-100 (1.9 mg/kg/day) or zanamivir (3.3 mg/kg/day) at 0-4, 2-6, 4-8, or 6-10 dpi. Bodyweight and survival were monitored daily for 14 days. **a** Changes in body weight and **b**, survival rate. Results are shown as mean values, and error bars indicate SD. **c–f** Efficacy of early and delayed administration of iHA-100. H5N1-infected mice were administered iHA-100 (1.9 mg/kg/day) (closed squares) or vehicle (open triangles) at 0-4 dpi (c, e; early administration) or 4-8 dpi (d, f; delayed administration), and sacrificed at 3, 5, or 7 dpi (c, e; early administration), or 5, 7, or 9 dpi (d, f; delayed administration; 4 dpi sacrifice was performed before administration). Lung weights in iHA-100-treated mice (closed squares) or vehicle-treated mice (open triangles) with early (**c**) and delayed (**d**) administration were measured and normalized to the body weight. Virus titers in the lungs of mice with early (**e**) and delayed (**f**) administration were determined with plaque assays in MDCK cells (detection limit: 250 PFU/g lungs). Results are shown as mean values, and error bars indicate SD. **g** Effect of iHA-100 on H5N1-induced pathogenesis. Representative histopathologic findings with H&E staining of the lungs of infected mice given early administration (0-4 dpi) of vehicle or iHA-100 at 3 and 7 dpi are shown. Scale bar = 100 μm.

more potent, broad-spectrum anti-influenza macrocycles will be developed in the near future.

## Methods

**Ethics statement.** All experiments using mice were approved by the Tokyo Metropolitan Institute of Medical Science Animal Experiment Committee and were performed in accordance with the animal experimentation guidelines of the Tokyo Metropolitan Institute of Medical Science. All macaque experiments were approved by the Shiga University of Medical Science Animal Experiment Committee and were performed in accordance with the Guidelines for the Husbandry and Management of Laboratory Animals of the Research Center for Animal Life Science at the Shiga University of Medical Science and with Fundamental Guidelines for Proper Conduct of Animal Experiment and Related Activities in Academic Research Institutions under the jurisdiction of the Ministry of Education, Culture, Sports, Science and Technology, Japan.

**Viruses and cells.** Madin-Darby canine kidney (MDCK) cells were maintained in Dulbecco's modified Eagle's medium supplemented with 10% fetal bovine serum, non-essential amino acids, sodium pyruvate, glucose, penicillin, and streptomycin. Human embryonic kidney 293 (HEK293) cells and the derivative HEK293T cells were maintained in Dulbecco's modified Eagle's medium supplemented with 10% fetal bovine serum, penicillin, and streptomycin. All cells were cultured at 37 °C in 5% $CO_2$.

Low pathogenic avian virus A/duck/Hokkaido/Vac-3/07 (H5N1; clade unclassified); highly pathogenic avian viruses A/Vietnam/UT3040/2004 (H5N1; clade 1), A/whooper swan/Mongolia/3/05 (H5N1; clade 2.2), and A/whooper swan/Hokkaido/1/08 (H5N1; clade 2.3); laboratory strains A/Puerto Rico/8/34 (H1N1), pandemic 2009 virus A/Tokyo/2619/09 (H1N1), and A/Narita/09 (H1N1); and A/Adachi/2/57 (H2N2) were used.

**In vitro peptide library selection.** Macrocyclic peptides were selected by Peptidream Inc. (Kanagawa, Japan). Baculovirus-derived recombinant full-length HA protein of A/Vietnam/1203/04 (H5N1; clade 1) (PROSPEC, East Brunswick, NJ) (iHA1-24) or vaccinia virus-derived recombinant full-length HA of A/Bar-Headed goose/Qinghai Lake/1 A/05 (iHA100-103), and 2.5 equivalents of Sulfo-NHS-LC-Biotin (Thermo Fisher Scientific, Waltham, MA) were incubated in an aqueous sodium phosphate solution (pH 8.0) at ambient temperature for 1 h. Biotinylated HA was immobilized on Dynabeads M-280 Streptavidin (Life Technologies, Carlsbad, CA). DNA templates and the codon table for peptide expression were constructed first[15]. For the initiation of translation, tRNAfMetCAU was charged with N-(2-chloroacetyl)-L- or D-tryptophan by flexizyme[11–13,26] and used in the cell-free translation system at a final concentration of 50 μM. The volume of the translation system for the first selection round was 200 μL, which could generate about $4 \times 10^{13}$ fusion molecules. The fusion molecules and the HA-immobilized beads were incubated in TBST (50 mM Tris-HCl buffer, pH 7.5, 120 mM NaCl, and 0.05% Tween20) at 4 °C for 1 h. The concentration of the immobilized protein was estimated at 250 nM in the incubation solution. The beads were washed with cold TBST three times and heated in TBST at 95 °C for 5 min. The supernatants were taken as eluates, and a fraction of the supernatant was subjected to real-time PCR for quantification of recovered fusion molecules. The remainder was used for preparative PCR. The PCR products were transcribed to generate templates for the next selection round. After five rounds of selection, in which the recovery rate of templates from the target-immobilized beads was about 50-fold higher than that from the control beads, the recovered templates were ligated to cloning plasmids (pGEM-T Easy; Promega, Madison, WI) for sequencing.

**Chemical synthesis of iHAs.** Macrocyclic peptides were synthesized by Peptidream Inc. All chemical peptides were assembled on Peptide Synthesizer SyroI (Biotage, Uppsala, Sweden) using the standard solid-phase peptide synthesis method with a 9-fluorenylmethoxycarbonyl (Fmoc) protecting group for the α-amino group of the amino acids. Rink Amide resin, 2-(1H-benzotriazol-1-yl)-

1,1,3,3-tetramethyluronium hexafluorophosphate, and Fmoc-amino acids were purchased from Novabiochem (Merck4Biosciences, Darmstadt, Germany). Piperidine, N,N-dimethylformamide, and 1-methyl-2-pyrrolidinone were purchased from Nacalai Tesque (Kyoto, Japan). N,N-diisopropylethylamine was purchased from Watanabe Chemical Industries (Hiroshima, Japan). After the last Fmoc group was removed, six equivalents of N-hydroxysuccinimide chloroacetate dissolved in 1-methyl-2-pyrrolidinone were added to the peptide resin and mixed for 1 h. The peptide resin was washed with dimethylformamide and dichloromethane three times each, dried, and then mixed with trifluoroacetic acid:triisopropylsilane:$H_2O$ (90:5:5, v/v/v) for 2 h. The cleaved peptide was precipitated by pouring cold tert-butyl methyl ether, washed with diethyl ether twice, and dissolved in dimethylsulfoxide (DMSO) so that the final concentration was about 10 mM. Six equivalents of triethylamine were added, and the mixture was stirred for 30 min at ambient temperature. The macrocyclic peptides were purified by reversed-phase chromatography with a C4 column Cosmosil AR300 5C4 (Nacalai Tesque). The purified peptides were identified by MALDI-TOF-MS measurements by using an AutoFlex II (Bruker, Billerica, MA).

The negative control iHA-4sf or iHA-12sf was synthesized based on the scrambled peptide sequence of peptide iHA-4 or iHA-12 (Fig. 1a). The iHA-4sf peptide sequence is Cyclo(HLSWELYDQYGWNC)G, and the iHA-12sf peptide sequence is Cyclo(VNWKFPLEWAIRYC)G.

**Plaque formation assay.** In some experiments (see Figure legends for details), viruses were pre-mixed with macrocyclic peptides at 37 °C for 60 min. MDCK cells in six-well plates were washed with minimum essential medium (MEM) twice, inoculated with 50–100 PFU of viruses (200 μL), and incubated at 37 °C for 60 min with rocking every 15 min. After removing the viruses, cells were washed with MEM and overlaid with agarose medium containing macrocyclic peptides selected in this study or zanamivir (GlaxoSmithKline, Middlesex, UK). After incubation of cells at 37 °C for 2–3 days, plaques were visualized with crystal violet staining and counted.

**Virus pull-down assay.** H5N1/Vac-3, H1N1/Narita, or H2N2/Adachi viruses ($1 \times 10^5$ PFU each) were mixed with biotin-iHAs (final concentration 1 μM) and incubated at 37 °C for 1 h. The virus-iHA mixtures were further incubated with NeutrAvidin-agarose (Pierce, Rockford, IL) at 4 °C for 2 h. After washing the agarose four times, eluates were subjected to RNA extraction with a QIAamp Viral RNA Mini Kit (Qiagen, Venlo, Netherlands). Viral RNAs from precipitated viruses were quantified for the M gene with the real-time thermal cycler CFX96 (Bio-Rad, Hercules, CA) and RNA-direct SYBR Green Realtime PCR Master Mix (Toyobo, Osaka, Japan)[27] using primers listed in Supplementary Table 2.

The effect of the biotinylated macrocyclic peptides on plaque formation due to H5N1/Vac-3 virus replication was evaluated with the plaque formation assay as described above. The negative control biotin-iHA-4sf was synthesized based on the sequence of peptide iHA-4 (Fig. 1b) and exhibited no inhibitory effect on virus replication. This negative control was composed of the scrambled peptide sequence: Cyclo(HLSWELYDQYGWNC)GG-polyethylene glycol (PEG)-Biotin.

**Polykaryon formation assay.** HEK293 cells ($5 \times 10^4$ cells/well) in 48-well plates were transfected with the protein expression plasmid pCAGGS for HA from A/Vietnam/1194/04 (H5N1; clade 1), A/Indonesia/5/05 (H5N1; clade 2.1), A/Bar-Headed goose/Qinghai Lake/1 A/05 (H5N1; clade 2.2), A/Anhui/1/05 (H5N1; clade 2.3), or A/California/07/2009 (H1N1). At 24 h post-transfection, cells were treated with acetylated trypsin (1 μg/mL) in MEM containing 0.3% bovine serum albumin at 37 °C for 30 min to cleave the HA into its HA1 and HA2 subunits and then treated with low pH buffer (MEM containing 20 mM sodium citrate (pH 5.0)) for 2 min at 37 °C. Cells were further incubated in a complete medium containing various concentrations of iHA-100 (0.01, 0.1, or 1 μM) at 37 °C for 6 h and then fixed with 4% paraformaldehyde. HA expression was detected with immunofluorescence staining with anti-HA rabbit polyclonal antibodies (Clones Pep10-(DAAEQTRLYQNPTTYISVG)-#11 for H5 and Pep20-(YSKKFKPEIAIRC)-#20

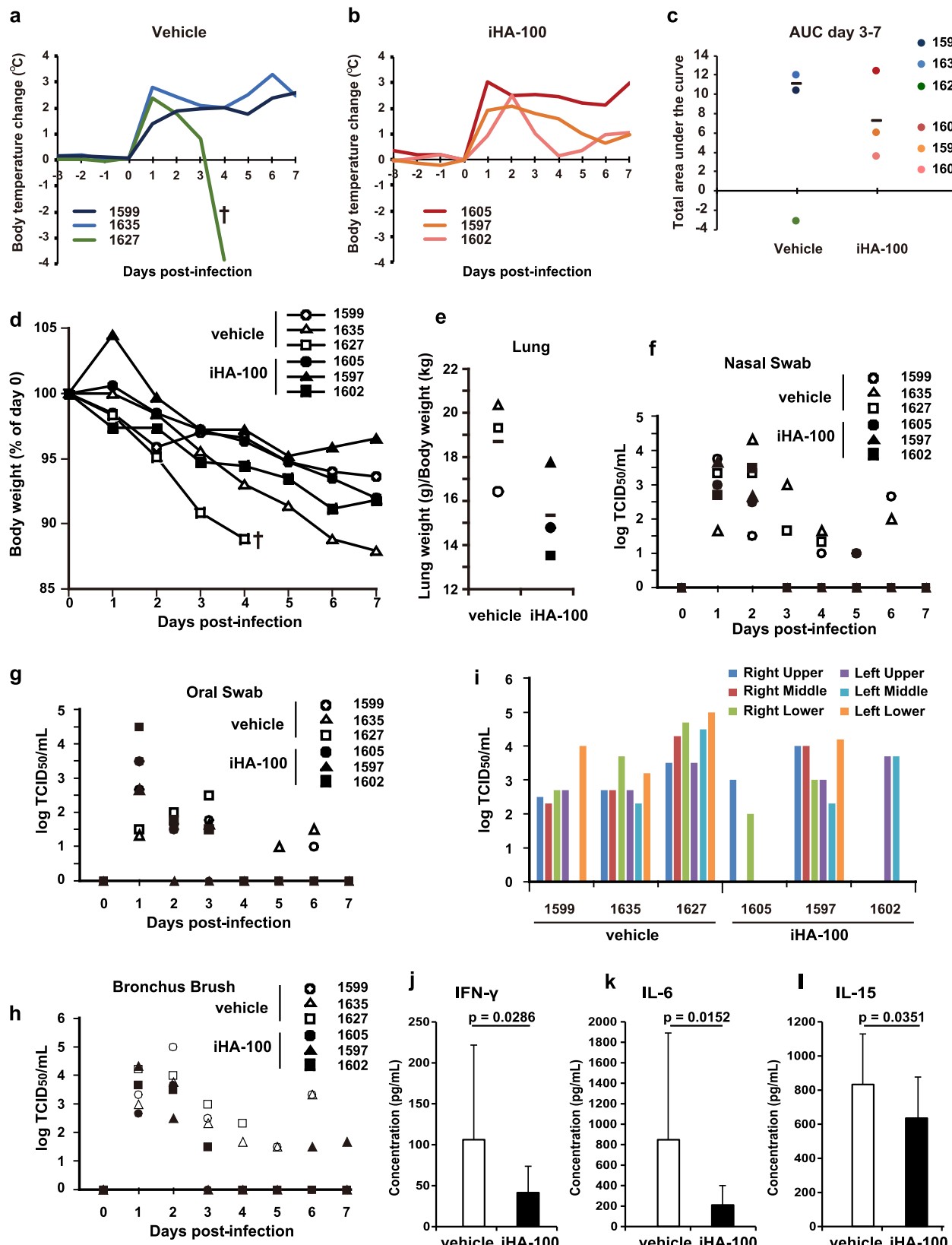

for H1 HA; prepared in our laboratory) and Alexa Fluor-488-conjugated secondary antibodies (Life Technologies). Nuclei were counterstained with TO-PRO-3 Iodide (Life Technologies). Cells were observed under a confocal laser microscope (FV300; Olympus, Tokyo, Japan). For quantification of the polykaryon formation efficiency, numbers of nuclei in five randomly chosen fields were divided by counterparts of the HA-expressing cells. Percentage (%) of polykaryon formation was normalized to the no treatment (NT) control.

**RBC fusion assay**. H5N1/Vac-3 or H1N1/pdm2619 virus ($1 \times 10^7$ PFU each) was incubated with chicken RBCs (final concentration 2%) on ice for 10 min. Dilutions of iHA-24 were added to the virus-RBC mixtures and incubated on ice for 30 min. The pH of the mixture was then reduced to 5.0 by adding sodium citrate buffer (pH 4.6), followed by further incubation at 37 °C for 60 min. After centrifugation at 3000 rpm ($800 \times g$) for 3 min, clarified supernatants were used for the determination of the NADPH concentration by measuring the optical density at 340 nm.

**Fig. 4 Evaluation of iHA-100 against influenza virus infection in a primate infection model.** Cynomolgus monkeys were intratracheally, intranasally, and orally infected with a total of $3 \times 10^6$ PFU/monkey of A/Vietnam/UT3040/2004 (H5N1/Vietnam) and were then intratracheally administered 3 mg/kg/day iHA-100 once daily for 7 days from 2 dpi. **a, b** Body temperature changes in monkeys in the vehicle-treated group (**a**) and iHA-100-treated group (**b**). **c** Summations of body temperature increase after treatment (AUC day 3 to day 7). Left, vehicle group; right, iHA-100 group. The bar symbol means the average of the vehicle group (except for ID #1627 because of low body temperature attributed to a serious clinical condition) and the iHA-100 group. **d** Bodyweight change in monkeys infected with influenza virus and treated with iHA-100 (closed symbols) or vehicle (open symbols). **e** Weights of lungs in vehicle-treated monkeys (open symbols) or iHA-100-treated monkeys (closed symbols) were measured and normalized to body weight. **f–h** Nasal, oral, and bronchial swabs were collected every day from each monkey and used to determine the virus titer with $TCID_{50}$. Open symbols indicate the titers of vehicle-treated monkeys. Closed symbols indicate the titers of iHA-100-treated monkeys. **i** Autopsies of H5N1-infected monkeys were performed 7 days after infection (the sample from #1627 was collected at 4 dpi). Each lobe of the lung was homogenized and used to determine virus titers. Results are shown as mean values and are representative of three biologically independent experiments ($n = 3$). **j–l** Comprehensive cytokine analysis of lungs removed from H5N1-infected monkeys. Each lobe of the lung was homogenized and used for the multiplex assay for 23-plex non-human primate cytokines. The six-lobe mixes (Right upper: RU, Right middle: RM, Right lower: RL, Left upper: LU, Left middle: LM and Left lower: LL) of the lung homogenates of each monkey were used for measurements, and the concentration is represented as the average of three monkeys in the vehicle (white bars)-treated or iHA-100 (black bars)-treated group. Results are shown as mean values and are representative of three biologically independent experiments. Vertical bars indicate the SD of three replicates. P values were calculated by Student's t-test (two-sided, unpaired).

**Isolation of influenza A virus escape mutants from iHA-100.** Ten-fold serial dilutions of H5N1/Vac-3 and H1N1/PR8 virus ($10^1$ to $10^8$ PFU) were mixed with iHA-100 (0.1 and 0.01 μM, respectively), and MDCK cells were inoculated with viral dilutions in a 96-well plate. After 3–4 days of incubation, the supernatants of the cytopathic effect-positive wells that were infected with the highest dilutions of virus (H5N1/Vac-3: $10^6$ PFU and H1N1/PR8: $10^4$ PFU) were subjected to plaque purification in six-well plates in a medium containing iHA-100 (0.1 and 0.01 μM). Ten virus clones each derived from H5N1/Vac-3 and H1N1/PR8 were selected for amplification in 10-cm dishes in a medium containing iHA-100 (0.1 μM). RNAs were extracted from the two plaque-purified viruses with the acid guanidinium thiocyanate-phenol-chloroform extraction method and subjected to reverse transcription using Superscript III reverse transcriptase (Life Technologies) for the influenza viral genome (all eight genes). The cDNAs for the HA gene were amplified with PCR and subjected to DNA sequencing using the BigDye Terminator 3.1 reagent and 3130 Genetic Analyzer (Life Technologies) using primers listed in Supplementary Table 2.

**Surface plasmon resonance (SPR).** All SPR experiments were performed with the Biacore X100 Plus Package (Cytiva, Tokyo, Japan) at 25 °C[28]. His-tagged HA was immobilized on a Sensor Chip NTA (Cytiva) activated by $Ni^{2+}$. iHA-100, dissolved in DMSO, was diluted in running buffer (1× phosphate-buffered saline (PBS), pH 7.4, 0.05% surfactant P20, and 2% DMSO). Binding constants were obtained from a series of injections of iHA-100 from 0.1 nM to 100 nM with a flow rate of 30 μL/min. Data from single-cycle kinetics were analyzed by BIAevaluation. The reference sensorgrams were subtracted from the experimental sensorgrams to produce curves representing specific binding, followed by background subtraction. Binding kinetics was evaluated using a 1:1 binding model to obtain association rate constants ($k_a$) and dissociation rate constants ($k_d$). Binding affinity ($K_D$) was calculated from those constants.

**Trypsin susceptibility assay.** For the trypsin susceptibility assay, 0.5 μM HA was pre-incubated at pH 8.0 with 2.5 μM iHA-100, 1.0 μM 14A7, or 1.0 μM CR6261 for 30 min at room temperature[18]. 14A7 is an HA head binding antibody produced in-house[29]. CR6261 is an HA stalk binding antibody that protects HA from pH-induced proteases[17]. Control reactions contained 2% DMSO without iHA-100. The pH of each reaction was lowered to pH 4.9 using 1 M sodium acetate buffer, while one reaction was retained at pH 8.0 to examine trypsin digestion at near-neutral pH. The reaction solutions were then thoroughly mixed by pipetting and incubated for 30 min at 37 °C. After incubation, the reaction solutions were neutralized to pH 8.0 with 1 M Tris-HCl buffer. Trypsin-ultra (New England Biolabs, Ipswich, MA) was added at a final ratio of 1:50 by mass, and the samples were digested for 20 min at 37 °C. Trypsin digestion was quenched by the addition of SDS loading buffer, and samples were boiled for 3 min at 99 °C. Finally, all samples were analyzed with 4–20% SDS-PAGE and immunoblotted with rabbit anti-H5HA antibody (Pep-10#11, produced in-house).

**Murine lethal infection model.** Five BALB/c female mice (8–9 weeks old) (Japan SLC, Hamamatsu, Japan) per group were anesthetized by injection of ketamine and then intranasally inoculated with 50% mouse lethal dose (5 $MLD_{50}$) of H5N1/Hokkaido virus (50 μL). Mice with bodyweight loss of more than 25% of their pre-infection value were euthanized. On 14 days post-infection, mice were euthanized, and the lungs and spleen were isolated. Histopathological analysis was performed with hematoxylin and eosin (H&E) staining, and virus titers in the lungs were determined with plaque assays in MDCK cells. For therapeutic treatment, H5N1/Hokkaido-infected mice were administered iHA-100 (1.9 mg/kg/day in Maintenance medium/0.4% PEG/0.4% 2-hydroxypropyl-beta-cyclodextrin (HPCD),

50 μL) or zanamivir (3.3 mg/kg/day in PBS, 50 μL) intranasally 3 h after infection and once daily for 4 days at 0–4, 2–6, 4–8, or 6–10 dpi. The vehicle for iHA-100 (Maintenance medium/0.4% PEG/0.4% HPCD, 50 μL) or zanamivir (PBS, 50 μL) was administered at 0–4 dpi. Bodyweights and survival of infected mice were monitored daily for 14 days. Mice with bodyweight loss of more than 25% of their pre-infection value were euthanized. On 3, 5, and 7 dpi (for early administration at 0–4 dpi) or 4, 5, 7, and 9 dpi (for delayed administration at 4–8 dpi), mice were euthanized, and the lungs and spleen were isolated. Histopathological analysis was performed with H&E staining, and virus titers in the lungs were determined with plaque assays in MDCK cells.

**Non-human primate infection model.** Three cynomolgus monkeys (3–4 years old, female) per group were anesthetized by injection of ketamine and xylazine and then intranasally (500 μL × 2), orally (500 μL × 2), and intratracheally (2.5 mL × 2) inoculated with a total of $3 \times 10^6$ PFU of H5N1/Vietnam. The infected monkeys were administered iHA-100 (3 mg/kg/day in Maintenance medium/0.74% PEG/0.74% HPCD) or vehicle (Maintenance medium/0.74% PEG/0.74% HPCD) intranasally (250 μL × 2), orally (250 μL × 2), and intratracheally (1 mL × 2) at 2–6 dpi. Bodyweight, body temperature, and clinical signs of infected monkeys were monitored daily for 7 days. Body temperature was monitored using telemetry. Body temperature changes of cynomolgus macaques were calculated as follows: Average temperatures of individual macaques from 8 p.m. to 8 a.m. every night were calculated from the temperature recorded using a telemetry probe (TA10CTA-D70, Data Sciences International, St. Paul, MN) every 15 min (e.g., averages on day −1 indicate from 8 p.m. on day −1 to 8 a.m. on day 0 before virus inoculation, and averages on day 0 indicate from 8 p.m. on day 0 to 8 a.m. on day 1 after virus inoculation). Body temperature changes in individual macaques on each day after virus inoculation were compared with the average body temperature from day 0 before virus inoculation. Swabs were sampled from the eye, nose, oral cavity, trachea, bronchus, and rectum of each monkey. These swabs were added to the MDCK cell culture for titration of the virus. At 7 dpi, monkeys were sacrificed under ketamine and pentobarbital anesthesia, and an autopsy was performed. Histopathological analysis was performed with H&E staining, and virus titers in the lung lobes were determined by $TCID_{50}$ in MDCK cells.

**Multiplex cytokine assay.** Comprehensive cytokine analysis of the lungs from H5N1-infected monkeys was performed using the Milliplex Map Non-Human Primate Cytokine Magnetic Bead Panel (Millipore, PCYTMG-40K-PX23) according to the manufacturer's instructions. Each lobe (RU, RM, RL, LU, LM, and LL) of the lung from the monkey was homogenized in lysis buffer (0.01 M Tris-HCl pH 7.5, 0.15 M NaCl, 1% Triton-X100, 20 mM EDTA) and used in the multiplex assay for 23-plex non-human primate cytokines.

**Serum stability assay.** iHA-100 (200 μM) and internal standard peptide (20 μM) (Supplementary Fig. 11, $NH_2$-$PEG_5$-$^DW^DS^DT^DN^DD^DW^DS^DT^DN^DD$-$PEG_5$-$CONH_2$) were co-incubated in human serum (Cosmo Bio) at 37°C for 0, 1, 3, 8, 24, and 48 h. Note that the internal standard peptide is peptidase resistant, and no degradation was observed under these experimental conditions. At each time point, 15 μL of the serum mixture was added to 35 μL methanol, incubated on ice for 5 min, and centrifuged at 13,000 rpm (15,300×g) at 25 °C for 3 min. The supernatant (30 μL) was mixed with 120 μL of 1% (v/v) trifluoroacetic acid and centrifuged at 13,000 rpm (15,300×g) at 25 °C for 3 min. Then, 2 μL of the supernatant was analyzed with liquid chromatography/mass spectrometry (LC/MS) (Xevo G2-XS QTof system, Waters) using a reverse-phase column (ACQUITY UPLC BEH C18, 1.7 μm, 2.1 × 150 mm, Waters) with a linear gradient from 1% buffer B to 60%

buffer B. Buffer B is CH$_3$CN containing 0.1% (v/v) formic acid, and buffer A is H$_2$O containing 0.1% (v/v) formic acid.

**Statistical analysis**. The significance of differences in the viral RNA titers in pull-down assays was assessed with the Dunnett's test and cytokine assay, lung weight and virus amout were assesed by two-tailed Student's *t*-test. To evaluate differences in other assays including plaque formation and polykaryon formation, one-way analysis of variance followed by Bonferroni's *post hoc* comparison tests were applied. *P* values <0.05 were considered statistically significant.

## Data availability

All relevant data are available from the corresponding author on request. The source data file is provided through figshare (https://doi.org/10.6084/m9.figshare.14363303).

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

## Acknowledgements

We thank all members of our laboratory for their advice and assistance. We are very grateful to Ms. Yoshimi Tobita, Dr. Keisuke Munekata, and Dr. Kousuke Saito for their technical support. This study was supported in part by grants from the Ministry of Education, Culture, Sports, Science and Technology of Japan (15H04739); Program for Promotion of Fundamental Studies in Pandemic Influenza of the Tokyo Metropolitan Government; the Ministry of Health, Labour and Welfare of Japan; and the Japan Initiative for Global Research Network on Infectious Diseases (J-GRID) from the Japan Agency for Medical Research and Development (AMED) (JP19fm0108006). This work was also supported by the Platform Project for Supporting Drug Discovery and Life Science Research (Basis for Supporting Innovative Drug Discovery and Life Science Research (BINDS)) from AMED under grant number JP19am0101090j0003 to H.S. The funders had no role in the study design, data collection, and analysis, decision to publish, or preparation of the manuscript.

## Author contributions

M.S., Y.I., F.Y., K.O., K.T.-K., H.S., and M.K. designed the study. M.S., Y.I., F.Y., T.M., D.Y., M.O., R.I., T.K., H.I., M.N., S.S., K.Y., N.Y., A.I., T.H., T.S., Y.S., H.K., L.T.Q.M., Y.K., and M.K. performed the experiments. M.S., Y.I., F.Y., T.M., D.Y., M.O., K.O., K.T.-K., H.S., and M.K. analyzed the data. M.S., Y.I., F.Y., Y.K., K.O., K.T.-K., H.S., and M.K. wrote the manuscript. All authors reviewed the manuscript.

## Competing interests

The authors declare no competing interests.
