## [Peer Review File · Nature Communications]

Reviewers' comments:

Reviewer #1 (Remarks to the Author):

Saito M et al. have reported the application of random non-standard peptides integrated discovery (RaPID) platform to design the potent macrocyclic peptides that target influenza virus hemagglutinin. The designed peptide neutralizes group 1 influenza virus by inhibiting hemagglutinin-mediated absorption and fusion. Further, the peptide is efficacious in the mouse and non-primate cynomolgus macaque models. Therefore, the reported peptide could be a lead to develop into an influenza therapeutic and the overall study should be of wide interest to the readers of this journal. However, certain issues need to be addressed.

1. Authors should report a brief description on the design and selection of peptides using the random non-standard peptides integrated discovery (RaPID) system reported in the manuscript.
2. In Figure 1a, it looks like peptide 11 is cyclic and rest of peptides are linear. Authors should report precise chemical linkage for rest of the peptides in the figure legend or may consider modifying figure 1a if all the peptides are indeed cyclic.
3. Figure 1c, rationale for methylation of the backbone of iHA-100 peptide is not reported in the manuscript. Do these modifications increase the metabolic stability compared to iHA-24 and rest of the designed peptides?
4. Throughout the manuscript, inhibitory or effective concentrations (IC₅₀ or EC₅₀) values seem to be missing for the peptides tested.
5. Binding of peptides are mostly reported using a whole virus assay. The authors should consider reporting direct inhibition of recombinant HA and reporting K_d values for at least the most potent peptide iHA-100.
6. The virus assays demonstrate that peptides inhibit the absorption and fusion stage of the virus life cycle (Figure 2). Further escape mutant indicate the peptide probably binds to the HA stem region (SI fig.6). The authors should comment on how peptide binding to the stem region rather than to the

head region (receptor binding site) inhibits the virus absorption to cells as this seems unexpected to me.

7. In absence of structural investigation, the exact mode of action of these peptides is unknown. The authors should consider also reporting the hemagglutination and trypsin digestion assays to gain the initial insight into HA receptor binding or probable stem binding of these peptides.

8. In Figure 3a, peptide iHA-100 shows more fluctuation in the percentage body weight compared to control? Also, on Page 9, lines 193-194. "loss was attenuated more in the monkeys in the iHA-100-treated group than in the vehicle-treated group" In 2 of 3 monkeys only?

9. Considering the good in vivo efficacy of these peptides in the mouse and non-primate cynomolgus macaque models, what is the metabolic stability in human plasma and off target effects of these peptides?

10. Page 7, lines 137-143. So how does this work?

11. Page 8, line 171. "no significant difference" Figure S6b says it is escape mutants – but they are all the same residues except for one and don't correspond to the escape mutants in Fig. S6a?

12. Page 8, line 171. There are no statistics presented so can't use "significant". Also the body weight of the iHA-100 treated group does seem to decrease and not completely recover compared to zanamivir at 0-4 dpi.

13. Page 19, lines 432-433, Reference 6. Incomplete reference.

14. Page 20, line 445, Reference 12. Please replace "(eds. Knipe, D.M., et al.) 1647-1689" with "(eds. Knipe, D.M., et al.) pp 1647-1689"

Reviewer #2 (Remarks to the Author):

The evaluation of anti-viral activity of iHA-100 in non-human primates is performed using an appropriate experimental setup and the appropriate highly pathogenic H5N1 virus. The routes of virus inoculation and dose used are in agreement with what is used in the field. However, the number of animals per study group is very low, making statistical evaluation difficult.

Comments:

1. Figure 4a shows a high and persistent increase in temperature in the vehicle treated animals that is consistent with the virus infection model. In the iHA-100 treated animals the temperature increase seems to be somewhat lower after day 4. However, this could be quantified further by subtracting the normal circadian temperature patterns and calculating the net increase (Mooij et al. 2015, PLoS ONE 10, e0126132).
2. Animals typically lose weight because of daily handling after study initiation. This makes it difficult to draw conclusions from a loss of weight. The differences between the groups are probably not significant.
3. The differences in lung weight are interesting, but do not reach significance.
4. Although the iHA-100 treated group appears to show lower virus shedding in nose and oral swabs and bronchus brush the results could be made clearer by calculating the area under the curve and performing statistical analysis on those results.
5. The method used for recording temperature should be mentioned in the methods section.

The presentation of the macaque data needs revision. I realize that adding more cynomolgus monkeys per group is difficult to realize. Therefore, the authors are asked to write in the discussion that the NHP data supports the mouse data and that the NHP part is more a proof of concept design.

Reviewers' comments:

Reviewer #1 (Remarks to the Author):

Saito M et al. have reported the application of random non-standard peptides integrated discovery (RaPID) platform to design the potent macrocyclic peptides that target influenza virus hemagglutinin. The designed peptide neutralizes group 1 influenza virus by inhibiting hemagglutinin-mediated absorption and fusion. Further, the peptide is efficacious in the mouse and non-primate cynomolgus macaque models. Therefore, the reported peptide could be a lead to develop into an influenza therapeutic and the overall study should be of wide interest to the readers of this journal. However, certain issues need to be addressed.

1. Authors should report a brief description on the design and selection of peptides using the random non-standard peptides integrated discovery (RaPID) system reported in the manuscript.

>To briefly describe the technology, we added a few sentences to describe the RaPID system as follows (page 6, lines 100–107):

To devise smaller molecules capable of binding to the influenza viral HA as potential antiviral agents (**Supplementary Fig. 1a**), we used an emerging technology called the Random non-standard Peptides Integrated Discovery (RaPID) system. This technology allowed us to express thioether-macrocyclic peptides in a custom-made *in vitro* translation system and display this massive library (greater than 10^{12} members) on cognate mRNA templates. Iterative selection rounds were then performed to enrich potent macrocyclic binders against recombinant HA derived from the highly pathogenic avian influenza virus A/Vietnam/1203/04 (H5N1/VietNam1203; clade 1) or A/Bar-Headed goose/Qinghai Lake/1A/05 (H5N1/Qinghai Lake: clade 2.2) (**Supplementary Fig. 1b**).

2. In Figure 1a, it looks like peptide 11 is cyclic and rest of peptides are linear. Authors should report precise chemical linkage for rest of the peptides in the figure legend or may consider modifying figure 1a if all the peptides are indeed cyclic.

>We apologize that this figure was not clear. They are all cyclic peptides, and we modified **Fig. 1a** and **Supplementary Fig. 1c** to clearly represent the macrocycle scaffold.

3. Figure 1c, rationale for methylation of the backbone of iHA-100 peptide is not reported in the manuscript. Do these modifications increase the metabolic stability compared to iHA-24 and rest of the designed peptides?

>We performed selections in a successive manner, i.e., we first performed selection using the thioether-macrocycle library consisting of proteinogenic amino acids in the random region, and then we performed selection using the thioether-macrocycle library containing four types of N-methyl-amino acids along with 11 proteinogenic amino acids in the random region. The latter library was designed to assure better metabolic stability than the former type of simple thioether-macrocytic peptides. In fact, new data were added to the revised manuscript, showing the serum stability of iHA-100 (**Supplementary Fig. 11**). We added the following sentences on page 6, lines 107–111.

In this study, we utilized two libraries of thioether-macrocytes in which the random region consists of only proteinogenic amino acids or both 11 proteinogenic and four types of N-methyl-amino acids of choice¹⁷. The latter library was designed to assure greater metabolic stability than the former type of simple thioether-macrocytic peptides.

4. Throughout the manuscript, inhibitory or effective concentrations (IC₅₀ or EC₅₀) values seem to be missing for the peptides tested.

> We appreciate the reviewer's comment. We have indicated the EC₅₀ values in **Fig. 1d-k**, **Supplementary Table 1**, and **Fig. 2g (right)**.

5. Binding of peptides are mostly reported using a whole virus assay. The authors should consider reporting direct inhibition of recombinant HA and reporting kd values for at least the most potent peptide iHA-100.

> We examined whether iHA-100 could block trypsin cleavage of recombinant HA0 protein using a trypsin susceptibility assay. Although HA0 was partially susceptible to trypsin under neutral pH conditions, HA0 was not digested by trypsin in the presence of iHA-100 under neutral pH conditions (**Fig. 2i, compare lanes 4 and 6**). These results indicate that iHA-100 had a protective effect on HA0 to prevent trypsin cleavage or digestion. We also performed a binding experiment using recombinant HA protein and determined $K_D = 1.5E-09$ with SPR. We described the results on p 7, lines 144-146, and in **Fig. 2b** and **Supplementary Fig. 4**.

6. The virus assays demonstrate that peptides inhibit the absorption and fusion stage of the virus life cycle (Figure 2). Further escape mutant indicates the peptide probably binds to the HA stem region (SI fig.6). The authors should comment on how peptide binding to the stem region rather than to the head region (receptor binding site) inhibits the virus absorption to cells as this seems unexpected to me.

>We appreciate the reviewer’s comment. We examined whether iHA-100 could block trypsin cleavage of HA0 using a trypsin susceptibility assay. HA was normally degraded by trypsin completely at acidic pH and partially at neutral pH, but when incubated with iHA-100 or CR6261, HA became protease resistant (**Fig. 2i**), suggesting binding of iHA-100 to the HA stalk region. These results also indicated that iHA-100 protected HA from trypsin cleavage at neutral pH (**Fig. 2i, compare lanes 4 and 6**). We consider that iHA-100 primarily interacted with the stalk domain to interfere with the conformational change in the HA protein at neutral conditions to block the viral adsorption process.

7. In absence of structural investigation, the exact mode of action of these peptides is unknown. The authors should consider also reporting the hemagglutination and trypsin digestion assays to gain the initial insight into HA receptor binding or probable stem binding of these peptides.

> We have found that iHA-100 could not inhibit hemagglutination in the Hemagglutination Inhibition Assay with chicken red blood cells (right figure). MAb 6F12 binds to the stalk domain of HA but does not have hemagglutination inhibition activity (Gene S. Tan et al. J. Virol. 2012;86:6179-6188). Thus, these results indicated that iHA-100 might target the stalk domain of HA.

We have found that iHA-100 protects HA from pH-induced trypsin digestion (**Fig. 2i**). Exposure to low pH renders H5 HA sensitive to trypsin digestion, but iHA-100 prevents conversion to the protease-susceptible conformation. 14A7 and CR6261 are anti-HA head and stem antibodies, respectively. CR6261 functions as a positive control in the trypsin susceptibility assays. These

results support the possibility that iHA-100 may bind the stem region of HA protein (page 8, lines 174–page 9, line 185).

8. In Figure 3a, peptide iHA-100 shows more fluctuation in the percentage body weight compared to control?

> Treatment of iHA-100 induced some body weight fluctuation when we administrated 0 day after virus infection (Fig.3a). The exact reason is still unclear at present. As reviewer pointed, our previous description “no significant difference was seen in body weight change” is not appropriate, therefore we have modified (p9, line 188-191).

Also, on Page 9, lines 193-194. “loss was attenuated more in the monkeys in the iHA-100-treated group than in the vehicle-treated group” In 2 of 3 monkeys only?

>In Fig. 4d, page 10, lines 215-216, we have modified the text based on your comments: “loss was relatively attenuated in the monkeys in the iHA-100-treated group than in two vehicle-treated monkeys (1627 and 1635)”.

9. Considering the good in vivo efficacy of these peptides in the mouse and non-primate cynomolgus macaque models, what is the metabolic stability in human plasma and off target effects of these peptides?

metabolic stability in human plasma:

>Based on this comment, we have performed an additional experiment to examine the stability of iHA-100 in serum (**Supplementary Fig. 11**). Note that the reason for the use of serum rather than plasma is that protease activity in plasma often varies and rapidly declines in an unfrozen state during incubation for a long period of time (>3.85 hours); therefore, we prefer to use serum for such a study. Despite the interesting question regarding off-target effects, defining off-target identities and their individual effects is difficult. Because we did not observe any cytotoxic effects using a control open peptide (vehicle, Fig. 3 and 4), we think that the peptide is indeed specific to HA, and we expect no strong off-target effects.

10. Page 7, lines 137-143. So how does this work?

>The suppressive effects of iHA-100 on influenza virus replication are likely caused by inhibition of the fusion step of influenza virus by binding to the stem region of HA. We have added this explanation (page 8, line 174–page 9, line 185).

11. Page 8, line 171. “no significant difference” Figure S6b says it is escape mutants – but they are all the same residues except for one and don’t correspond to the escape mutants in Fig. S6a?

>As the reviewer pointed out, Figure S6b was confusing due to a lack of information. Therefore, we now show a new **Fig. 2h** and new **Supplementary Fig. 7a**.

12. Page 8, line 171. There are no statistics presented so can’t use “significant”. Also the body weight of the iHA-100 treated group does seem to decrease and not completely recover compared to zanamivir at 0-4 dpi.

>We have modified the text, based on the reviewer’s comment (page 9, lines 188–191).

13. Page 19, lines 432-433, Reference 6. Incomplete reference.

>We have corrected reference 6 (page 21, line 498-499).

14. Page 20, line 445, Reference 12. Please replace “(eds. Knipe, D.M., et al.) 1647-1689” with “(eds. Knipe, D.M., et al.) pp 1647-1689”

>We have modified the reference based on the reviewer’s comment (page 21, line 510-512).

Reviewer #2 (Remarks to the Author):

Review manuscript Saito et al. Nature Communications NCOMMS-20-00265-T

The evaluation of anti-viral activity of iHA-100 in non-human primates is performed using an appropriate experimental setup and the appropriate highly pathogenic H5N1 virus. The routes of virus inoculation and dose used are in agreement with what is used in the field. However, the number of animals per study group is very low, making statistical evaluation difficult.

>We agree with the reviewer that the number of animals per study group is low. However, we could not perform this study using more animals in our facility. We have used three monkeys (**Fig. 4**) in each experimental group, and therefore, we can perform statistical evaluation.

Comments:

1. Figure 4a shows a high and persistent increase in temperature in the vehicle treated animals that is consistent with the virus infection model. In the iHA-100 treated animals the temperature increase seems to be somewhat lower after day 4. However, this could be quantified further by subtracting the normal circadian temperature patterns and calculating the net increase (Mooij et al. 2015, PLoS ONE 10, e0126132).

>We appreciate the reviewer's comment. To express body temperature changes quantitatively, differences in body temperature before and after infection were calculated in the revised **Fig. 4a** and **b**. For comparison, averages of total body temperature increase in macaques are shown as the AUC in the revised **Fig. 4c**. The AUC of body temperature increases after treatment with iHA-100 was lower than that of body temperature increases after treatment with the vehicle (page 10, line 211-212).

2. Animals typically lose weight because of daily handling after study initiation. This makes it difficult to draw conclusions from a loss of weight. The differences between the groups are probably not significant.

>We started daily handling at least 1 weeks before the start of the experiment, and the beginning of the experiment did not significantly influence the decrease in body weight ($p > 0.05$) (**Fig. 3a** and **Fig. 4**). The difference in body weight between the experimental groups in **Fig. 3a** and **Fig. 4d** was not significant within 1-2 days after infection.

3. The differences in lung weight are interesting, but do not reach significance.

>Based on the statistical analysis, the effect of iHA-100 in preventing a pneumonia-associated increase in lung weight was obvious but not significant ($p = 0.059$, one-tailed statistical analysis) (page 10, line 218).

4. Although the iHA-100 treated group appears to show lower virus shedding in nose and oral swabs and bronchus brush the results could be made clearer by calculating the area under the curve and performing statistical analysis on those results.

>Thank you very much for your constructive comments. We have calculated the area under the curve and performed statistical analysis on the results in **Fig. 4f** (Nasal swab), **g** (Oral swab), and **h** (Bronchus swab), which showed statistical significance ($p = 0.042$, 0.021 and 0.016 , respectively). The area under the curve for the iHA-100-treated group showed significantly lower virus shedding in nasal swabs, oral swabs, and bronchus brush (**Supplementary Fig. 8**) (page 10, line 220-222).

5. The method used for recording temperature should be mentioned in the methods section.

>Body temperature was measured using telemetry and was described in the text (page 19, line 442-447).

6.The presentation of the macaque data needs revision. I realize that adding more cynomolgus monkeys per group is difficult to realize. Therefore, the authors are asked to write in the discussion that the NHP data supports the mouse data and that the NHP part is more a proof of concept design.

>Thank you very much for your constructive comments. We have added the requested information, based on the reviewer's comment (page 11, lines 235–237).

REVIEWERS' COMMENTS

Reviewer #1 (Remarks to the Author):

The authors have addressed all of my comments in the previous review and have updated the revised manuscript accordingly. Thus, the revised manuscript is now suitable for publication.

Reviewer #2 (Remarks to the Author):

My original criticisms and comments have been answered and the authors have improved their manuscript.